# Determination of Strain Properties of the Leather Semi-Finished Product and Moisture-Removing Materials of Compression Rolls

**DOI:** 10.3390/ma12213620

**Published:** 2019-11-04

**Authors:** Auezhan Amanov, Gayrat Bahadirov, Tileubay Amanov, Gerasim Tsoy, Ayder Nabiev

**Affiliations:** 1Department of Mechanical Engineering, Sun Moon University, Asan 31460, Korea; 2Institute of Mechanics and Seismic Stability of Structures, Academy of Sciences, Tashkent 100125, Uzbekistan; instmech@rambler.ru (G.B.); amanov43@mail.ru (T.A.); tsoygeran@mail.ru (G.T.); a.nabiev@mail.ru (A.N.)

**Keywords:** leather semi-finished product, compression rolls, strain properties, topographic sections, moisture extraction

## Abstract

The paper presents the results of experimental studies to determine the strain properties and characteristics of a chrome leather semi-finished product of middle-weight bovine hide by its topographic sections and the coatings of the processing compression rolls. The strain pressure equations of depend on the topographic sections of a leather semi-finished product are obtained, and the results of experiments on the effect of a number of layers on the amount of pressed moisture are presented. A mathematical dependence of the pressed moisture from the leather semi-finished product is obtained under various pressure values, rates, and the number of skin layers with monchons. The influence of the number of layers of leather semi-finished products with moisture-removing materials (monchons) on the process of moisture extraction at their vertical feed on the base plate is determined. In this paper, the influence of the number of skin layers and moisture-removing materials (monchons) on the technological process of moisture extraction from wet leather semi-finished products at their vertical feed on a base plate is studied as well.

## 1. Introduction

It is known that the leather semi-finished product has a complex natural structure, which differs in species, breed, sex, age of the animal, as well as by topographic sections and other factors. The technology of mechanical processing of a leather semi-finished product is influenced by various external and internal factors associated with its physical, mechanical, and strain properties. Studies of the properties of various leather semi-finished products were carried out in several investigations. The fibrous structure of skin presents a great difficulty in studying the patterns of skin strain, as different types of skin have different thickness and softness, in different directions and different topographic sections [1,2,3,4,5,6,7,8,9].

In [10], technological defects, which are often encountered during leather semi-finished product compression, were considered, and the measures to eliminate them were described. The importance of the necessary pressure of compression rolls, the moisture content in leather semi-finished products, the feed rate of the latter into the treatment zone, and many other factors were noted in that paper.

In this work, we have experimentally determined the strain properties and characteristics of a chrome leather semi-finished product of middle-weight bovine skin in topographic sections and coatings of processing working rolls. It should be noted that the samples of leather semi-finished products used in the experiments were produced in Uzbekistan.

In Uzbekistan, more than 94% of the total number of cattle is kept and raised in private farms and households. Livestock farms raise mainly such cattle breeds as Black-motley, Red-Steppe, Shvits, Bushuevskaya, Santa Gertrude, and local breeds. Constant, purposeful work is underway to improve livestock keep and the quality of hides. As a result of modern veterinary services for livestock farms, a favorable epizootic situation has been created in the country. In all categories of farms, the cattle livestock is increasing every year. More than nine million hides are processed annually at the tannery enterprises of the republic. The obtained results will be used in the design and development of technological machines for leather production, and the selection of rational modes of operation of technological machines and tannery equipment. This will improve the quality of individual technological operations, and ultimately the quality of the finished leather.

The uneven surface and thickness of the leather semi-finished product in all the topographic sections create considerable difficulty in selecting the optimal elastic characteristics of the working rolls coating, and do not provide a uniform residual humidity of 55–60% in the topographic sections of the leather semi-finished product. To eliminate this drawback in the technology of moisture extraction from the leather semi-finished product, a lying stage is provided after the extraction; it takes a certain time, after which, as a rule, the process proceeds to the next technological operation. To eliminate the lying of the leather semi-finished product, it is necessary to study the strain properties in the topographic sections of the leather semi-finished product to ensure the uniformity of residual moisture and thereby to shorten the time for the lying. Therefore, it is necessary to take into account the changes in the strain properties of the leather semi-finished product in its topographic sections when calculating and designing technological machines for leather working. For this purpose, a device was designed and manufactured to measure the strain characteristics of the leather semi-finished product under load.

## 2. Experimental Procedures and Results

An average weight bovine hide was taken in the experiment as the material of the leather semi-finished product after chrome tanning and bifurcation. Chrome tanning is the process of treating skins to produce leather, which uses a solution of chromium sulfate to tan the hide. In turn, bifurcation is the process of splitting of a one-piece leather into sections (see Figure 1). According to the state standard GOST 938075, for the experiment a certain number of leather semi-finished products was selected according to the formula n=0.2√x, where n is the number of leather semi-finished products in the experiment, and x is the number of leather semi-finished products in the batch. At x = 2500 pcs, n = 10 pcs. From these 10 leather semi-finished products, the thickness was measured in three points in three topographic sections of skin—butt, shoulder, and belly section—and the average thickness was determined (Table 1, Figure 1). The thickness of the leather semi-finished product was measured after the liquid operation of tanning and bedding before the technological operation of squeezing the liquid. The thickness of the leather semi-finished product was measured with a thickness gauge in which the pressure of the moving platform on the leather semi-finished product was laid on a fixed platform and created using a thickness gauge spring. By pressing the lever of the thickness gauge, the movable platform was lifted, a leather semi-finished product was placed between the measuring platforms, and the lever was smoothly lowered. Then, the indication of the dial arrow was noted, and thus the thickness of the leather semi-finished product was set. The measurements were carried out with an absolute error of up to 0.01 mm. To measure the thickness of the leather semi-finished product by topographic section, the pressure of the upper pad of the thickness gauge on the leather semi-finished product was 49 kPa, since the pressure did not cause noticeable deformation of the leather semi-finished product at the time of measurement.

Then, the strain ε in the leather semi-finished product was determined on the stand depending on the stress *σ*. The measurements were carried out with a micrometer of a watch type with an accuracy of 0.01 mm. As shown in Table 2, strains were measured at stress values of 2.5 × 10^5^ Pa, 5 × 10^5^ Pa, 10 × 10^5^ Pa, 20 × 10^5^ Pa, 30 × 10^5^ Pa, 40 × 10^5^ Pa, 50 × 10^5^ Pa in the butt, shoulder, and belly. The strains in the moisture-removing materials of the cloth of БМ and ЛАЩ brands caused by compression pressure were measured and listed in Table 2. The graphs of the strain–compression pressure dependence were plotted (Figure 2).

The influence of number of skin layers and moisture-removing materials (monchons) on the technological process of moisture extraction from wet leather semi-finished products was measured at their vertical feed on a base plate. The experiment was carried out on a special stand, where the compression rolls were installed horizontally, and the base plate was made of a metal sheet with a thickness of 0.005 m, a width of 0.1 m, and a length of 0.3 m. The scheme of wet leather semi-finished products fed to the extraction zone is shown in Figure 3.

One layer of the sandwich consists of one leather semi-finished product followed by two moisture-removing materials (monchons)—the LASCH cloth, and so on. A device to measure the strain characteristics of the materials consists of a plate on which a rod is mounted, a frame is fixed on the rod, and an indicator is fixed on the frame. On the frame, the guides are installed and fixed on top with nuts to the plate, as well as fixed at the bottom to the plate. A load is installed on the plate. A cylinder is installed on the plate, and a leather semi-finished product is installed on the cylinder. The plate presses the cylinder, which is installed on the leather semi-finished product.

Equations of strain–compression pressure dependences are obtained:

1. For a leather semi-finished product (butt):(1)$$\sigma_{bt}=-4.4449+15.9306\cdot \varepsilon -6.6352\cdot \varepsilon_{}^{2}+8.8864\cdot \varepsilon^{3}$$

2. For a leather semi-finished product (shoulder):(2)$$\sigma_{sh}=-3.5789-0.6979\cdot \varepsilon_{}+13.0072\cdot \varepsilon_{}^{2}-2.7189\cdot \varepsilon^{3}$$

3. For a leather semi-finished product (belly):(3)$$\sigma_{bl}=-124.6234+286.3747\cdot \varepsilon -204.6262\cdot \varepsilon_{}^{2}+49.2680\cdot \varepsilon^{3}$$

4. For a moisture-removing material, cloth BM:(4)$$\sigma_{BM}=-28.4232+62.5300\cdot \varepsilon -40.3361\cdot \varepsilon_{}^{2}-9.2408\cdot \varepsilon^{3}$$

5. For a moisture-removing material, cloth LASCH:(5)$$\sigma_{LASCH}-99.0366+121.8463\cdot \varepsilon_{}-49.4131\cdot \varepsilon_{}^{2}+6.7761\cdot \varepsilon^{3}$$

According to experimental tests, the dependence of stress σ on strain ε was determined for the butt, shoulder, belly, and for moisture-removing materials of BM and LASCH brands from Table 3.

The analysis of experimental data on strain in the leather semi-finished product from the compression pressure shows that the strain fluctuations are from 46.6% at a compression pressure of 2.5 × 10^5^ Pa to 23.7% at a compression pressure of 50 × 10^5^ Pa. The mathematical dependences of strain in the leather semi-finished product on the compression pressure in the topographic sections in the butt, shoulder, belly, and moisture-removing materials in the cloth of BM and LASCH brands are obtained. It was established that in the initial loading section, the strain in the samples of the leather semi-finished product and coatings of the working rolls increases sharply, and then the dependence takes a linear character. It should be noted that the samples of the chrome leather semi-finished product were of local origin and procurement to be able to identify the factors affecting the quality of the finished leather. The obtained equations of the strain–pressure dependence are used to improve the calculation method and in the design of newly developed technological devices and machines for leather working, taking into consideration the influence of such factors as the number of leather layers with moisture-removing materials (monchons) on the process of extracting moisture from wet leather at a vertical feeding on the base plate. A patent was obtained for an invention of a method of moisture extraction from skins at a multiple sandwich with monchons [11]. The experiment was carried out on a special stand (see Figure 4a), where the compression rolls were installed horizontally, and the base plate was made of a metal sheet 0.005 m thick, 0.1 m wide, and 0.3 m long. A detailed view of the experimental stand in Figure 4b contains a mechanism for controlling the intensity of the clamping force, which helps provides the necessary pressure values of the squeeze shafts P = 32 kN/m, P = 64 kN/m, and P = 96 kN/m by tightening the springs with a piston with a rod. The feed rate of the multilayer package into the extraction zone was provided by chain transmission of the necessary speed, and revolutions of the squeeze shafts V = 0.17 m/s, V = 0.255 m/s, V = 0.34 m/s were regulated by a slide rheostat current of a DC motor with a power of 3 kW. One layer of a leather sandwich consists of one skin and one moisture-removing material (monchon)—a LASH cloth.

An average weight bovine hide was taken in the experiment as the leather material after chrome tanning and bifurcation. According to state standards GOST 938075, for the experiment, a certain number of leather semi-finished products was selected according to the formula n = 0.2√x, where n is the number of skins in the experiment, and x is the number of skins in the batch. At x = 2500 pcs, n = 10 pcs. From these 10 skins, the strips were cut with a cutter across the spine line of 0.05 × 0.25 m in size; the strips were numbered and combined in groups of 40 sets of two pieces, 25 sets of five pieces, and 40 sets of eight pieces. A total of 53 strips were cut out from one skin. A total of 530 pieces [12] was tested by the method of asymmetric fringe.

The experiment was carried out as follows:

A strip of LASCH fiber cloth 0.004-m thick was mounted on a metal base plate, followed by leather, and so on. Then, they turned on the stand, set the springs compression calibration to the desired pressure, the speed was regulated by a rheostat, and the rotating speed of the rolls was regulated by tachometer ТЧ-10P. Control skin samples were preliminarily fed and the spring compression was measured, i.e., the deviation from the first set. If the deviation exceeded 3%, the springs were adjusted by tightening the nuts. Then, the basic skin samples were fed. The samples were weighed on a laboratory scale, ВЛТЭ-500, with a discreteness of 0.1 g (ISO-9001), before and after compression.

In the experimental results processing, the second-order D-optimal planning method was taken using the Kano plan matrix, since its application provides the highest accuracy in estimating the regression coefficients [13]. It was taken into account that the Kano plan provides for the factors variation at three levels: lower (−), zero (0), and upper (+), which is appropriate for this study. Based on a priori information, the process of moisture removal was studied taking into account three factors: *x*_1_—compression intensity P, kN/m; *x*_2_—feed rate V, m/s; and N_1_—the number of layers of the leather semi-finished product with monchon pieces (pcs). The range of the compression pressure was selected from 32 to 96 kN/m; the velocity of compression rolls was selected from 0.17 to 0.34 m/s; and the number of skin layers N was selected from 2 to 8 pcs, based on the analysis of various wringing machines produced in various countries as shown in Table 4. In the study, the diameter of the compression rolls was 0.2 m with a coating thickness 0.01 m of the cloth BM.

Before the experiment, the required number of measurements (a number of repeatability), which provided the required accuracy was selected by mathematical statistics methods.

The working matrix was compiled using the Kano plan matrix for a three-factor experiment. The coding of factors was carried out according to the formula:(6)$$x_{i}=\frac{c_{i}-c_{i0}}{t_{0}}$$
where *x_i_* is the coding of the factor value; *c_i_*, *c_i_*_0_ are the natural values of the factor at the current and zero levels; and *t*_0_ is the natural value of the interval of factor variation.

Target functions are approximated by a polynomial:(7)$$y=b_{0}+\sum_{i=1}^{k}b_{i}x_{i}+\sum_{i,j=1}^{k}b_{ij}x_{i}x_{j}+\sum_{i=1}^{k}b_{ii}x_{i}^{2}$$
where *у* is the amount of moisture removed in a coded form; *b*_0_, *b_i_, b_ij_,* and *b_ii_* are the regression coefficients.

After the implementation of the working matrix, arithmetic mean values were obtained (Table 5). The dispersion homogeneity was carried out using the Cochrane criterion with a confidence probability of α = 0.95.
(8)$$G_{cal}=\frac{S_{\max}^{2}}{\sum_{1}^{N}S_{i}^{2}}\leq G_{t};\;S_{er}^{2}=\frac{\sum_{1}^{n}{\left(y_{i}-{\bar{y}}_{i}\right)}^{2}}{n-1};\;G_{cal}=\frac{13.2857}{74.325}=0.1788$$

*G_cal_* = 0.1788 < *G_t_* = 0.192, N is the total number of variances, *y_i_* is a series of parallel experiments, ${\bar{y}}_{i}$ is the average value of each parallel experiment, and n is the number of parallel experiments.

Since *G*_cal_, the calculated value of the Cochrane criterion, is less than the tabular *G_t_*, the experiments are reproducible. We determine the regression coefficients *b*_0_, *b*_i_, *b*_ij_, and *b*_ii_ from the table in [12]: *b*_0_ = 20.8; *b*_11_ = −0.3616; *b*_22_ = 0.1113; *b*_33_ = 2.3923; *b*_1_ = 3.0543; *b*_2_ = −1.9025; *b*_3_ = −3.7802; *b*_12_ = −0.3154; *b*_13_ = −0.3014; and *b*_23_ = −0.1863.

The regression equation in a coded form is:(9)$$\begin{array}{l} y=20.8-0.3616\cdot x_{1}^{2}+0.113\cdot x_{2}^{2}+2.3923\cdot x_{3}^{2}+3.0543\cdot x_{1}-1.9025\cdot x_{2}-\\ -3.7802\cdot x_{3}-0.3154\cdot x_{1}\cdot x_{2}-0.3014\cdot x_{1}\cdot x_{3}-0.1863\cdot x_{2}\cdot x_{3}. \end{array}$$

The hypothesis of the adequacy of the obtained equations was checked using the Fisher’s test with the confidence probability α = 0.95:(10)$$F_{cal}=\frac{S_{ad}^{2}}{S^{2}\left\{у\right\}}≺F_{t}$$

$S_{ad}^{2}$ is the residual variance, or adequacy variance; $S^{2}\left\{y\right\}$ is the reproducibility dispersion.

From Table 1 and Table 2, $S_{ad}^{2}$ and $S^{2}\left\{y\right\}$ are defined:(11)$$S_{ad}^{2}=\frac{\sum_{1}^{N}n\cdot \left({\bar{y}}_{i}-{\hat{y}}_{i}\right)}{N-\frac{\left(k+2)(k+1\right)}{2}}=\frac{\sum_{1}^{21}5\cdot {\left({\bar{y}}_{i}-{\hat{y}}_{i}\right)}^{2}}{21-\frac{\left(3+2)(3+1\right)}{2}}=\frac{5\cdot 8.9281}{11}=4.0582$$
(12)$$S^{2}\left\{y\right\}=\frac{\sum_{1}^{N}\sum_{1}^{n}{\left(y_{i}-{\bar{y}}_{i}\right)}^{2}}{N(n-1)}=\frac{297.3002}{21(5-1)}=\frac{297.3002}{84}=3.5393$$

Fisher’s test for model adequacy is:(13)$$F_{cal}=\frac{S_{ad}^{2}}{S^{2}\left\{y\right\}}=\frac{4.0582}{3.5393}=1.1466.$$
$S_{ad}^{2}$ is the variance of adequacy; $S^{2}\left\{y\right\}$ is the reproducibility dispersion; *N* is the total number of experiments; *k* is the number of factors; *n* is the number of repetitions in the experiment; *у_i_* is the result of a separate observation; ${\bar{y}}_{i}$ is the arithmetic mean value of the result of experiment; ${\hat{y}}_{i}$ is the calculated values of the criterion according to the regression equation for $S_{ad}^{2}$ = 4.2491; $S^{2}\left\{y\right\}$ = 7.0078; *F_t_* = 2.40; *f*_1_ = 11; and *f*_2_ = 84. Since *F_cal_* = 1.1466 < *F_t_* = 2.40, we accept the model adequacy with a confidence probability of 0.95. The dependence of the removed moisture amount on the number of layers of a leather semi-finished product between compression rolls at a compression intensity is shown in Figure 5. The results of the measurement before and after the moisture extraction from the leather semi-finished product at various values of the pressing forces *x*_1_ (P) and wringing speed *x*_2_ (V) are shown in Table 6.

After working matrix implementation, the arithmetic mean values were obtained (Table 7).

The dispersion was homogeneous using the Cochrane criterion [12] with a confidence probability of *α* = 0.95. Knowing the total number of variance estimates *N* and the number of degrees of freedom *f = k–*1, from [12], we find *G*_t_ = 0.358, at *N* = 9, *f = k−*1 = 5–1 = 4.

*k* is the number of parallel experiments.
(14)$$S_{er}{}^{2}=\frac{\sum_{1}^{n}{\left(y-\bar{y}\right)}^{2}}{n-1};\;\sum_{1}^{N}S_{i}^{2}=\frac{\sum_{1}^{N}\sum_{1}^{n}{\left(y-\bar{y}\right)}^{2}}{N(n-1)}$$
(15)$$G_{cal}{}_{1}=\frac{S_{\max}^{2}}{\sum_{1}^{N}S_{i}^{2}}=\frac{13.84}{64.36}=0.215;\;G_{cal}{}_{2}=\frac{S_{\max}^{2}}{\sum_{1}^{N}S_{i}^{2}}=\frac{8.94}{34.74}=0.257$$

*G*_cal1_ = 0.215 < *G_t_* = 0.3584; *G*_cal2_ = 0.257 < *G_t_* = 0.38. Therefore, the results of the study are reproducible.

We determine the regression coefficients *b*_0_, *b_i_*, *b_ij_*, and *b_ii_* from Table 7 and Table 8. For the first leather semi-finished product in coded form, *b*_0_ = 20.9552; *b*_11_ = 0.5657; *b*_1_ = 5.567; *b*_22_ = 0.4157; *b*_2_ = −4.1363; and *b*_12_ = −0.925. For the second leather semi-finished product in coded form *b*_0_ = 21.7707; *b*_11_ = 0.0372; *b*_1_ = 5.2217; *b*_22_ = 0.3872; *b*_2_ = −3.0699; and *b*_12_ = −0.7. The following regression equations are obtained in coded form: 

For the first leather semi-finished product:(16)$$y_{1}=20.952+0.5657\cdot x_{1}^{2}+5.567\cdot x_{1}+0.4157\cdot x_{2}^{2}-4.1363\cdot x_{1}-0.925\cdot x_{1}x_{2}$$

For the second leather semi-finished product:(17)$$y_{2}=21.7707+0.0372\cdot x_{1}^{2}+5.2217\cdot x_{1}+0.3872\cdot x_{2}^{2}-3.0699\cdot x_{2}-0.7\cdot x_{1}x_{2}$$

Entering $x_{1}=\frac{P-64}{32}$, where P is the pressing force of compression rolls and $x_{2}=\frac{V-0.255}{0.085}$, where V is the feed rate of wet leather semi-finished products between rotating compression rolls, we got the equation of moisture amount removed from the wet leather semi-finished product, in percent, depending on the pressing force and the feed rate of the wet leather semi-finished product between the rotating compression rolls.

The hypothesis of the adequacy of the obtained equations was checked using Fisher’s test with the confidence probability α = 0.95 [12]:(18)$$F_{cal}=\frac{S_{ad}^{2}}{S^{2}\left\{у\right\}}≺F_{t}$$
where $S_{ad}^{2}$ is the residual variance, or adequacy variance; $S^{2}\left\{y\right\}$ is the reproducibility dispersion.

For the first leather semi-finished product:(19)$$S_{ad}^{2}=\frac{\sum_{1}^{N}n\cdot {\left(\bar{y}-y_{cal}\right)}^{2}}{N-\frac{\left(k+2)(k+1\right)}{2}}=\frac{5\cdot 6.36}{3}=10.6;\;S^{2}\left\{y\right\}=\frac{\sum_{1}^{N}\sum_{1}^{n}{\left(y-\bar{y}\right)}^{2}}{N(n-1)}=\frac{257.43}{9(5-1)}=7.445$$

For the second leather semi-finished product:(20)$$S_{ad}^{2}=\frac{\sum_{1}^{N}n\cdot {\left(\bar{y}-y_{cal}\right)}^{2}}{N-\frac{\left(k+2)(k+1\right)}{2}}=\frac{5\cdot 5.47}{3}=9.117$$
(21)$$S^{2}\left\{y\right\}=\frac{\sum_{1}^{N}\sum_{1}^{n}{\left(y-{\bar{y}}_{}\right)}^{2}}{N(n-1)}=\frac{138.94}{9(5-1)}=3.859.$$

Fisher’s test on model adequacy:(22)$$F_{cal}=\frac{S_{ad}^{2}}{S^{2}\left\{y\right\}}=\frac{4.0582}{3.5393}=1.1466$$
where *N* is the total number of experiments; *k* is the number of factors; *n* is the number of repetitions in the experiment; *у*_i_ is the result of a separate observation; $\bar{y}$ is the arithmetic mean values of the experiment result; and $y_{cal}$ is the calculated values of the criterion from the regression equation.

Therefore, the regression equation can be considered suitable with a 95% confidence probability, which in the nominated form after decoding is:

For the first layer of the leather semi-finished product:*y*_1_ = 22.8776 + 0.0006·*P*^2^ + 0.839·*P* + 57.4542·*V*^2^ – 56.2041·*V* – 0.34·*P*·*V*;(23)

For the second layer of the leather semi-finished product:*y*_2_ = 12.7039 + 0.000036·*P*^2^ + 0.2335·*P –* 53.5917·*V*^2^ + 7.6889·*V* – 0.2574·*P*·*V*.(24)

Graphs of dependence of the removed moisture W from the wet leather semi-finished product in percent were plotted for different feed rates V and compression forces P (see Figure 6). The results of experiments show that the difference in the moisture removed from the first and the second layers of a leather semi-finished product is insignificant. With a decrease in the pressing force of the compression rolls *P*, the difference in the removed moisture W between the first and second layers of the processed skins decreases. For a test sample of a leather semi-finished product taken from middle-weight bovine hide for finished upper shoes leathers, the maximum moisture content in the belly reaches 73%, and in the butt, it reaches up to 65%. Residual moisture in the leather semi-finished product after extraction is 55–60%, depending on the type of skin. In our case, the residual moisture should be about 60%. Therefore, we need to remove a maximum of 13% moisture content under squeezing on a roll machine. It follows that it is possible to squeeze out moisture from a wet leather semi-finished product at a feed rate of 0.34 m/s and with a pressing force of compression rolls from 32 to 96 kN/m. The experimental results showed that it is possible to simultaneously squeeze out wet leather semi-finished products and remove excess moisture from two leather semi-finished products (by layers) at the same time; this allows a 200% increase in the productivity of roller machines, which enables a reduction in energy costs to extract moisture from wet leather semi-finished products.

## 3. Conclusions

Thus, the regression equation can be considered suitable with a 95% confidence probability. The regression equation after coding will be as follows:(25)$$\begin{array}{c} \Delta W\;=\;20.2719-3.531\cdot 10^{-4}\cdot P^{2}\;+\;15.6401\cdot V{\;}^{2}\;+\;0.2658N_{1}^{2}\;+\;0.1859P\;-\;3.4129\cdot V-\\ 1.0106\cdot N_{1}\;-0.116\cdot P\cdot V\;-\;3.14\cdot 10^{-3}\;P\cdot N_{1}\;–\;0.7306\cdot V\cdot N_{1} \end{array}$$

An analysis of experimental results shows that with an increase in the extraction pressure, the amount of removed moisture increased, and with an increase in the extraction speed, the amount of removed moisture decreased.

Experimental results showed that it is possible to simultaneously squeeze out wet leather semi-finished products and remove excess moisture from two leather semi-finished products (by layers) at the same time; this allows a 200% increase in the productivity of roller machines, and thus a reduction in energy costs to extract moisture from wet leather semi-finished products.

## Figures and Tables

**Figure 1 materials-12-03620-f001:**
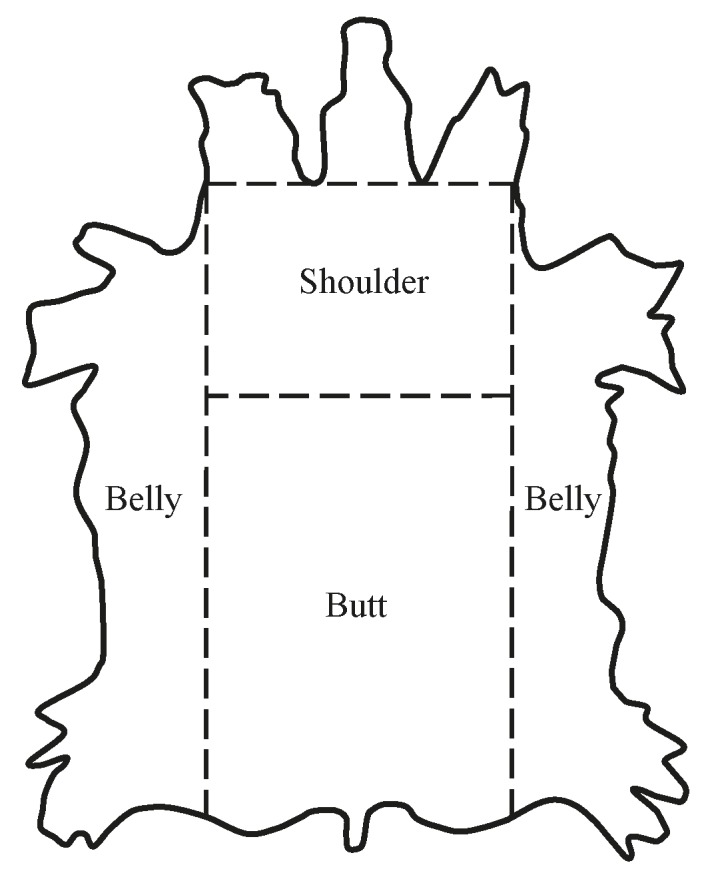
Topographic sections of a one-piece leather semi-finished product.

**Figure 2 materials-12-03620-f002:**
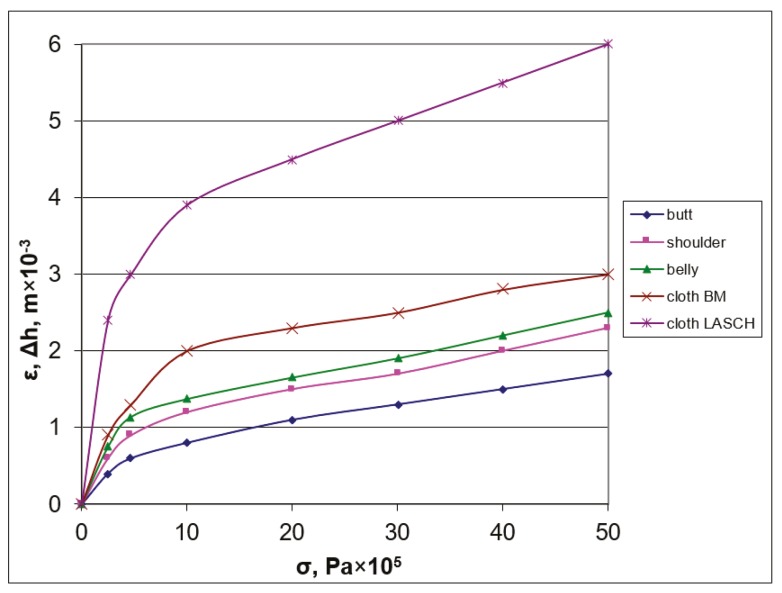
Strain dependence in the leather semi-finished product in topographic sections and moisture-removing materials on the compression pressure of the working roll.

**Figure 3 materials-12-03620-f003:**
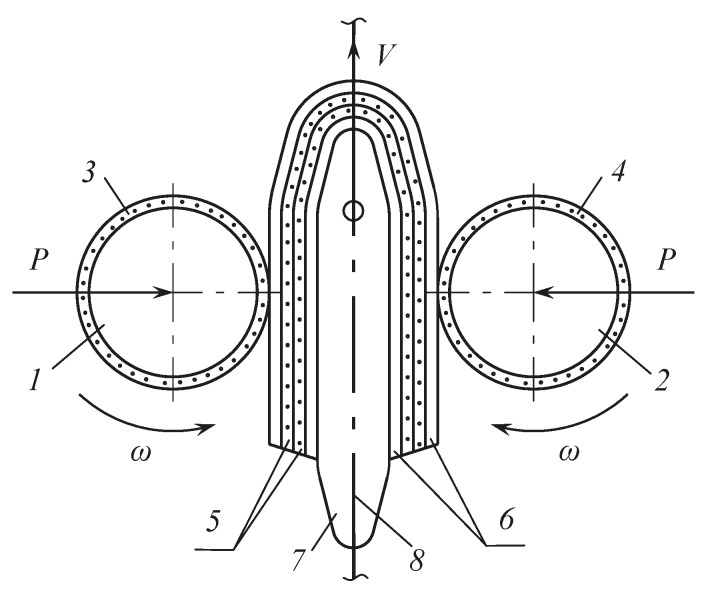
Schematic of wet leather semi-finished products fed to the extraction zone. 1, 2—compression rolls, 3, 4, 5—moisture-removing materials, 6—leather semi-finished products, 7—base plate, 8—drag chain.

**Figure 4 materials-12-03620-f004:**
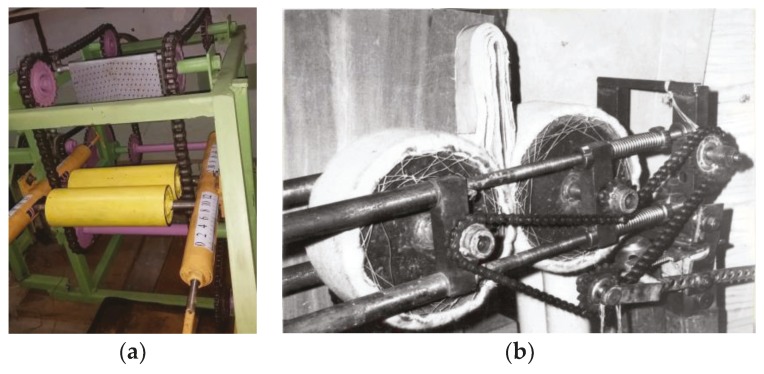
A stand for wringing moisture from a multiple skin sandwich with the vertical feed on the base plate (**a**). A detailed view of the stand (**b**).

**Figure 5 materials-12-03620-f005:**
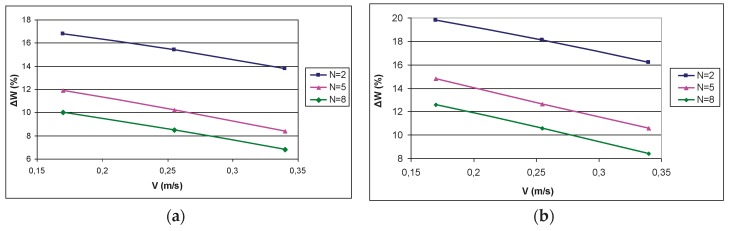
Dependence of removed moisture amount ΔW on the number of layers N of a leather semi-finished product between compression rolls at a compression intensity: P = 32 kN/m (**a**), P = 64 kN/m (**b**), P = 96 kN/m (**c**); at the leather semi-finished product thickness N = 2 pieces (pcs), N = 5 pcs, and N = 8 pcs.

**Figure 6 materials-12-03620-f006:**
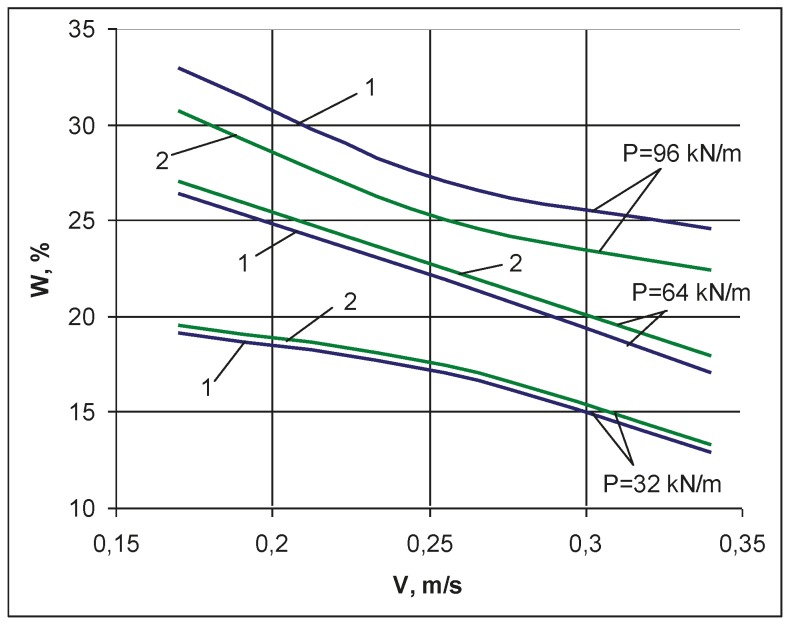
Dependence of the amount of removed moisture *(W)* on the feed rate *(V)* of the leather semi-finished product between compression rolls at an intensity of pressing force: P = 32 kN/m, P = 64 kN/m, P = 96 kN/m. 1—first layer of the leather semi-finished product; 2—second layer of the leather semi-finished product.

**Table 1 materials-12-03620-t001:** Thickness of the leather semi-finished products in topographic sections after tanning and bifurcation.

Semi-Finished Products	Thickness of the Leather Semi-Finished Product	Topographic Sections of One-Piece Leather Semi-Finished Product								
		Butt			Shoulder			Belly		
		Points			Points			Points		
		1	2	3	1	2	3	1	2	3
1	Average, 10^−3^ m	3.58	3.49	3.56	3.62	3.56	3.57	3.62	3.59	3.42
2	Average of Three Points, 10^−3^ m	-	3.54	-	-	3.58	-	-	3.54	-

**Table 2 materials-12-03620-t002:** Average strain in the materials from the force of compression pressure.

No.	Type of Material	Average Thickness, 10^−3^ m	Compression Pressure, 10^5^ Pa						
			2.5	5.0	10.0	20.0	30.0	40.0	50.0
1	Semi-Finished Leather Product (butt), 10^−3^ m	3.54	0.47	0.62	0.86	1.11	1.33	1.57	1.74
2	Semi-Finished Leather (shoulder), 10^−3^ m	3.58	0.78	0.94	1.22	1.48	1.73	2.0	2.24
3	Semi-Finished Leather (belly), 10^−3^ m	3.54	0.88	0.95	1.28	1.63	1.81	2.19	2.28
4	Cloth БM, 10^−3^ m	8	0.95	1.30	2.06	2.46	2.68	2.81	3.06
5	Cloth ЛАЩ, 10^−3^ m	6.8	2.43	3.03	3.44	3.58	3.76	3.96	4.3

**Table 3 materials-12-03620-t003:** Strains in leather semi-finished product in topographic sections from the compression pressure.

Type of Material	Compression Pressure on Leather Semi-Finished Product, 10^5^ Pa						
	2.5	5.0	10.0	20.0	30.0	40.0	50.0
Semi-Finished Leather Product (butt), 10^−3^ m	0.47	0.62	0.86	1.11	1.33	1.57	1.74
Semi-Finished Leather Product (shoulder), 10^−3^ m	0.78	0.94	1.22	1.48	1.73	2.0	2.24
Semi-Finished Leather Product (belly), 10^−3^ m	0.88	0.95	1.28	1.63	1.81	2.19	2.28
Maximum Fluctuation of Strain in Leather Semi-Finished Product from the Compression Pressure, 10^−3^ m	0.41	0.33	0.42	0.52	0.48	0.62	0.54
Maximum Fluctuation of Strain in Leather Semi-Finished Product from the Compression Pressure, %	46.6	31.4	32.8	31.9	26.5	28.3	0.24

**Table 4 materials-12-03620-t004:** Levels and ranges of factor variation in the experiment.

Index	Coded Values of Factors	Natural Values of Factors		
		*x*_1_, kN/m	*x*_2_, m/s	*x*_3_, pcs.
Upper Level	+	96	0.340	8
Zero Level	0	64	0.255	5
Lower Level	−	32	0.170	2
Variation Range		32	0.085	3

**Table 5 materials-12-03620-t005:** Plan matrix.

No.	*P* *x* _1_	*V* *x* _2_	*N_1_* *x* _3_	Measurements Results, in %						$\sum_{1}^{n}{\left(y-{\bar{y}}_{mean}\right)}^{2}$	$S_{er}^{2}$	$\hat{y}$	${\bar{y}}_{mean}-\hat{y}$	${\left({\bar{y}}_{mean}-\hat{y}\right)}^{2}$
				*у* _1_	*у* _2_	*у* _3_	*у* _4_	*у* _5_	${\bar{y}}_{mean}$					
1	2	3	4	5	6	7	8	9	10	11	12	13	14	15
1	0	0	0	22.49	21.38	20.90	20.81	21.88	21.50	1.4044	0.3511	20.81	0.69	0.4761
2	+	+	+	19.42	21.32	19.74	20.46	22.77	20.75	7.29	1.8225	19.51	1.24	1.5376
3	+	–	+	24.01	25.47	22.34	24.19	24.19	24.04	4.9808	1.2452	24.29	–0.25	0.0625
4	–	–	+	18.62	19.67	18.65	19.67	15.90	18.51	6.8121	1.7031	18.22	0.29	0.0841
5	–	+	+	14.97	13.36	12.11	15.89	14.17	14.10	8.4736	2.1184	14.66	–0.56	0.3136
6	+	+	–	27.44	27.63	31.26	25.03	27.15	27.71	25.0714	6.2679	28.05	–0.34	0.1149
7	+	–	–	33.96	31.97	33.12	34.66	30.41	32.83	11.3059	2.8265	32.05	0.33	0.1089
8	–	–	–	27.93	23.47	25.95	24.27	22.31	24.79	9.5108	2.3777	24.77	–0.02	0.0004
9	–	+	–	21.02	21.84	22.00	20.74	22.06	21.69	0.7069	0.1767	21.98	0.29	0.0841
10	+	0	+	21.63	20.57	22.07	20.07	20.82	21.14	2.6192	0.6548	22.02	–0.88	0.7744
11	0	–	+	21.08	22.33	20.65	21.93	17.77	20.76	13.7266	3.4317	20.54	–0.22	0.0484
12	–	0	+	14.66	17.85	16.18	19.52	16.13	16.88	13.4063	3.3516	16.51	0.31	0.0961
13	+	–	0	28.21	27.52	27.01	23.15	26.40	26.41	15.4152	3.3538	26.75	–0.29	0.0841
14	–	–	0	19.89	21.40	18.71	20.80	22.20	20.40	7.5162	1.8779	19.82	0.56	0.3136
15	–	+	0	15.46	17.75	16.03	17.83	16.43	16.70	4.3868	1.0967	16.51	0.19	0.0361
16	+	+	0	21.38	17.05	21.33	20.77	22.47	20.60	17.2696	4.3174	22.12	1.52	2.3104
17	0	+	+	18.72	16.78	16.08	17.02	13.54	16.43	14.1893	3.5474	16.50	–0.07	0.0049
18	+	0	–	30.95	29.06	34.58	27.84	31.63	30.82	26.7886	6.6972	30.18	0.64	0.4096
19	0	–	–	24.04	30.98	22.55	25.06	29.56	26.44	53.1425	13.2857	27.83	–1.39	1.9321
20	–	0	–	18.65	26.75	22.29	19.90	25.04	22.53	46.1374	11.5344	23.50	0.03	0.0009
21	0	+	–	25.77	26.16	26.28	26.21	24.52	24.79	7.1467	1.7867	24.43	0.36	0.1296
										297.3002	74.3250			8.9281

**Table 6 materials-12-03620-t006:** Experimental data on the moisture extraction from a wet leather semi-finished product.

No.	*x* _1_	*x* _2_	*y_1_, gr*		*y_2_, gr*		*y_3_, gr*		*y_4_, gr*		*y_5_, gr*	
			*y_in1_*	*y_fin1_*	*y_in2_*	*y_fin2_*	*y_in3_*	*y_fin3_*	*y_in4_*	*y_fin4_*	*y_in5_*	*y_fin5_*
1	+	–	70.5	48.6	77.0	54.3	94.0	63.8	84.2	52.6	86.6	56.7
			74.3	51.9	85.3	57.0	96.8	70.0	95.2	61.6	86.7	63.0
2	+	0	86.7	63.1	80.0	58.6	75.1	54.6	95.8	70.9	71.7	52.4
			98.3	72.8	62.5	45.2	83.3	61.5	86.7	67.2	79.4	59.1
3	+	+	68.3	53.8	81.6	60.8	84.5	62.5	79.3	60.1	73.3	58.3
			83.1	62.7	63.9	43.3	81.2	64.1	90.7	69.3	75.8	58.8
4	0	–	94.1	68.5	98.5	72.8	78.1	55.8	72.4	57.5	89.5	65.4
			91.6	62.3	66.9	45.5	85.1	63.6	86.9	71.3	92.0	65.3
5	0	0	88.7	66.0	93.1	72.3	76.2	62.7	88.4	65.9	68.4	56.7
			89.6	66.2	85.5	66.4	69.9	54.5	82.5	66.7	73.4	56.6
6	0	+	76.0	63.3	72.5	60.0	62.5	51.8	73.1	61.7	94.9	78.8
			75.5	61.7	66.1	54.4	94.9	78.2	68.1	55.7	92.5	76.1
7	–	–	92.6	77.2	97.5	78.6	81.2	62.6	80.2	55.5	92.6	76.0
			73.5	59.2	96.7	80.2	75.1	60.3	87.7	72.7	70.0	53.8
8	–	0	65.5	54.8	89.7	74.1	73.4	60.7	81.0	64.8	89.4	75.5
			73.5	60.1	79.7	66.3	73.0	59.5	84.8	72.4	76.1	63.2
9	–	+	66.0	55.3	62.5	54.4	91.9	81.4	70.7	60.0	84.3	73.7
			89.8	72.1	69.6	60.0	79.8	68.8	84.7	76.4	70.7	60.6

*y_in_* is the initial weight of the wet leather semi-finished product sample; *y_fin_* is the weight of the leather semi-finished product sample after extraction.

**Table 7 materials-12-03620-t007:** Plan matrix.

No.	*P*, *x*_1_	*V*, *x*_2_	Leather Semi-Finished Product	Measurements Results, in %						$\sum_{1}^{n}{\left(y-\bar{y}\right)}^{2}$	$S_{er}^{2}$	$y_{cal}$	$\bar{y}-y_{cal}$	${\left(\bar{y}-y_{cal}\right)}^{2}$
				*у* _1_	*у* _2_	*у* _3_	*у* _4_	*у* _5_	$\bar{y}$					
1	+	–	1	31.1	29.5	32.0	37.5	34.5	32.9	39.46	9.86	32.6	1.5	2.25
			2	30.2	33.2	27.7	35.3	27.3	30.7	35.75	8.94	29.7	1.0	1.0
2	+	0	1	27.2	26.8	27.3	26.9	26.9	27.0	0.15	0.04	27.2	0.2	0.04
			2	25.9	27.7	26.2	22.5	25.3	25.6	14.47	3.82	27.0	1.4	1.96
3	+	–	1	21.2	25.5	26.0	24.2	20.7	23.5	23.87	5.97	22.5	1.0	1.0
			2	24.5	24.4	21.1	23.6	22.4	23.2	8.36	2.08	22.8	0.4	0.16
4	0	–	1	27.2	26.1	30.9	20.6	27.0	26.4	54.98	13.74	5.5	0.9	0.81
			2	30.9	32.0	25.3	18.0	29.0	27.0	24.3	6.08	24.5	0.2	0.04
5	0	0	1	25.6	22.3	17.7	25.5	18.6	21.9	55.34	13.84	21.2	0.3	0.09
			2	26.1	22.3	22.0	19.2	22.9	22.5	24.3	6.08	21.9	0.6	0.36
6	0	+	1	16.8	17.2	17.1	16.9	17.0	17.0	0.10	0.25	16.9	0.6	0.36
			2	18.3	17.7	17.6	18.2	17.7	17.9	0.42	0.11	18.4	0.5	0.25
7	–	–	1	16.6	19.5	22.9	13.3	17.9	19.0	54.92	13.73	19.6	0.6	0.36
			2	19.5	17.1	19.7	18.2	21.1	19.1	9.33	2.33	18.2	0.9	0.81
8	–	–	1	16.3	17.4	17.3	20.0	14.6	17.2	15.46	3.87	16.0	1.2	1.44
			2	18.2	16.8	18.5	14.6	17.0	17.0	9.49	2.37	16.2	0.8	0.64
9	–	+	1	16.2	13.0	4.4	12.3	13.5	13.3	13.15	3.28	13.2	0.1	0.01
			2	13.0	13.3	13.8	9.8	14.3	12.9	12.54	3.13	13.4	0.5	0.25
										257.43	64.36			6.36
										138.94	34.74			5.47

**Table 8 materials-12-03620-t008:** Coefficient factors.

No.	*P*, *x*_1_	*V*, *x*_2_	Coefficients Factors						Leather Semi-Finished Product	$\bar{y}$
			*b* _0_	*b* _11_	*b* _22_	*b* _1_	*b* _2_	*b* _12_		
1	0	0	0.5772	−0.3234	−0.3234	0	0	0	1	21.9
									2	22.5
2	+	+	−0.1067	0.1691	0.1691	0.1961	0.1961	0.25	1	23.5
									2	23.2
3	–	+	−0.1067	0.1691	0.1691	−0.1961	0.1961	−0.25	1	13.3
									2	12.9
4	–	–	−0.1067	0.1691	0.1691	−0.1961	0.1961	0.25	1	19.0
									2	19.1
5	+	–	−0.1067	0.1691	0.1691	0.1961	−0.1961	−0.25	1	32.9
									2	30.7
6	+	0	0.2114	−0.3883	−0.3383	0.1078	0	0	1	25.0
									2	25.6
7	0	+	0.2114	−0.3383	0.1617	0	0.1078	0	1	17.0
									2	17.9
8	0	+	0.2114	0.1617	−0.3383	−0.1078	0	0	1	17.2
									2	17.0
9	–	0	0.2114	−0.3383	0.1617	0	−0.1078	0	1	26.4

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
