# Peer review of "Determination of Strain Properties of the Leather Semi-Finished Product and Moisture-Removing Materials of Compression Rolls"

_materials, 2019, doi:10.3390/ma12213620_

Round 1
Reviewer 1 Report
In this paper, a device was designed and manufactured to measure the strain characteristics of the leather semi-finished product under load.
The leather has a certain rebound after being compressed, so the author should give specific measurement methods and conditions for the thickness of the leather. The author should verify the resulting formula. The stand showed in Figure 4 is not detailed enough to explain how to feed and wind the multiple skin sandwich. Line 294, "With an increase in the number of layers of a leather semi-finished Product with a monchon, the amount of moisture removed from the leather semi-finished product Reduced." The author does not need to explain common sense after the trial.Author Response
Please see the attachment

Reviewer 2 Report
The authors in their paper describe the determination of Strain Properties of the Leather Semi-finished Product and Moisture-removing Materials of Compression Rolls.
We believe this is a very interesting argument and approach to gain deeper insight into the mechanism of leather processing. The work is nicely done and in good agreement with the scope of Materials journal. In our opinion the work is publishable in its current form.
Author Response
Thank you for your invaluable comments and opinions. We appreciate your time and effort to review the paper.
Reviewer 3 Report
In section 2. Experimental procedures and results, line 69 it should be explained better what means bifurcation (chrome tanning and bifurcation).
Author Response
In section 2. Experimental procedures and results, line 69 it should be explained better what means bifurcation (chrome tanning and bifurcation).
Response: Thank you for your comment. The following sentences have been added to the revised manuscript:
Chrome tanning is the process of treating skins to produce leather which uses a solution of chromium sulfate to tan the hide. In turn, bifurcation is the process of splitting of a one-piece leather into sections (see Fig. 1).